# Usability of Indocyanine Green in Robot-Assisted Hepatic Surgery

**DOI:** 10.3390/jcm10030456

**Published:** 2021-01-25

**Authors:** Anne-Sophie Mehdorn, Jan Henrik Beckmann, Felix Braun, Thomas Becker, Jan-Hendrik Egberts

**Affiliations:** Department of General, Abdominal, Thoracic, Transplantation and Pediatric Surgery, University Hospital Schleswig-Holstein, Campus Kiel, Arnold-Heller-Straße 3, 24105 Kiel, Germany; anne-sophie.mehdorn@uksh.de (A.-S.M.); jan.beckmann@uksh.de (J.H.B.); Felix.Braun@uksh.de (F.B.); Thomas.Becker@uksh.de (T.B.)

**Keywords:** robotic surgery, indocyanine green, robotic liver resection, da Vinci, intraoperative imaging, hepatocellular cancer, real-life imaging, hepatic metastasis

## Abstract

Recent developments in robotic surgery have led to an increasing number of robot-assisted hepatobiliary procedures. However, a limitation of robotic surgery is the missing haptic feedback. The fluorescent dye indocyanine green (ICG) may help in this context, which accumulates in hepatocellular cancers and around hepatic metastasis. ICG accumulation may be visualized by a near-infrared camera integrated into some robotic systems, helping to perform surgery more accurately. We aimed to test the feasibility of preoperative ICG application and its intraoperative use in patients suffering from hepatocellular carcinoma and metastasis of colorectal cancer, but also of other origins. In a single-arm, single-center feasibility study, we tested preoperative ICG application and its intraoperative use in patients undergoing robot-assisted hepatic resections. Twenty patients were included in the final analysis. ICG staining helped in most cases by detecting a clear lesion or additional metastases or when performing an R0 resection. However, it has limitations if applied too late before surgery and in patients suffering from severe liver cirrhosis. ICG staining may serve as a beneficial intraoperative aid in patients undergoing robot-assisted hepatic surgery. Dose and time of application and standardized fluorescence intensity need to be further determined.

## 1. Introduction

Hepatobiliary surgery has made great technological progress over time, developing from open surgery to minimally-invasive approaches including laparoscopy and, more recently, robot-assisted procedures [1,2,3,4]. Advantages of the robotic platform compared to laparoscopic surgery include smaller incisions, clearer visualization of structures, higher degrees of freedom, avoidance of the fulcrum effect, and better access to segments IVa, VII, and VIII [1,2,3,4,5]. However, robot-assisted liver surgery is still partly in its infancy and hepatic resections have not yet been standardized [1,4,5,6]. Additionally, either some instruments are unavailable for the robotic platform or they are difficult to handle or change [1,6,7]. With the development of more suitable devices and surgical instruments, robot-assisted liver surgery will further improve [1,8]. Nonetheless, until now robot-assisted liver surgery has been proven to be safe and comparable to open surgery and laparoscopy from both a surgical and oncological point of view [1,5,9,10,11,12,13].

However, one major drawback of minimally-invasive surgery is the lack of haptic feedback, since palpation with laparoscopic or robotic forceps is limited [2,3,5,12,14,15]. The surgeon must rely on their own visual impressions, making parenchymal dissections particularly problematic [3,7]. As an additional intraoperative aid for strategic and intraoperative planning, the water-soluble dye indocyanine green (ICG) has been suggested [16,17,18]. ICG was approved by the Food and Drug Administration (FDA) in 1957 and has been used in various medical fields [19,20,21]. Since the 1980s, ICG has been used to test liver function prior to hepatobiliary surgery. In this indication (LiMON test), ICG is administered intravenously days before surgery and the blood concentration and ICG plasma disappearance rates are measured noninvasively [14,22]. In healthy liver tissue, ICG is fully excreted after 72 h and no remnants should be detectable [1,14]. In this context, Ishizawa et al. noticed ICG accumulation in hepatocellular carcinoma (HCC) and hepatic metastasis (HM) of colorectal cancer up to 14 days after ICG application for liver function evaluation [16].

ICG accumulation can be visualized using a near-infrared (NIR) camera, which is currently often integrated into laparoscopic or robotic systems. The Firefly™ camera (Intuitive, Sunnyvale, CA, USA) is integrated in the da Vinci Surgical Systems (Intuitive, Sunnyvale, CA, USA) and can easily be used to intraoperatively visualize ICG accumulation [12]. In open surgery, additional NIR cameras and/or monitors are needed for ICG visualization, theater lights need to be switched off, and the operating surgeon must remove their focus from the operation field while performing crucial parts of the operation. In laparoscopic surgery, the NIR camera is integrated into some systems. Additionally, the quality of images highly depends on the systems used [15]. This results in a high risk for agitation within the theater at a vulnerable phase of surgery. Using the integrated Firefly^TM^ camera, the surgeon operating using the robotic system can continue to focus on the operation field because the Firefly^TM^ camera can easily be switched on and off and acquire real-life intraoperative ICG-based images that are directly projected onto the operation field. These may further be merged with intraoperative ultrasound (IOUS) images without changing instruments or monitors [7,12,23].

However, ICG is still mainly used to test liver function prior to hepatobiliary surgery or intraoperative bile leakage; it is not yet routinely applied to detect tumors or metastases [1]. Furthermore, reports on ICG-based tumor or metastatic resections mainly include open and laparoscopic procedures [12,15,16,19,23,24,25,26,27,28]. To our knowledge, only one group has reported on ICG-based, robot-assisted hepatic surgery for HCC and metastasis of colorectal cancer (CRC) [17,18]. Thus, we aimed to review our experiences regarding robot-assisted hepatic resections after ICG application in patients suffering from HCC and metastasis of CRC. Additionally, we wanted to evaluate the feasibility of preoperative ICG application in patients suffering from hepatic metastasis other than CRC metastasis.

## 2. Material and Methods

### 2.1. Data Collection

A prospective database collecting data for all patients undergoing robot-assisted hepatic surgery is maintained at the Dept. of General, Visceral, Transplant, Thoracic, and Pediatric Surgery, University Hospital Schleswig-Holstein, Campus Kiel, Germany.

Demographic data and the clinical courses of included patients were prospectively retrieved from the hospital’s in-house patient files. All patients provided written informed consent for inclusion in the study and use of their data. The local ethics committee provided written approval (D 610/20). The study adheres to the principles of the Declarations of Helsinki and Istanbul. Only de-identified data were used for further analysis. Patient data included age, gender, and body mass index (BMI), as well as tumor- and surgery-specific details.

### 2.2. Inclusion and Exclusion Criteria

The study included patients with primary HCC and HM of different origins, i.e., breast cancer, esophageal cancer, choroid coat melanoma, or neuroendocrine tumors, who were scheduled for robot-assisted hepatic liver resections between February 2019 and October 2020 at the Dept. of General, Visceral, Thoracic, Transplantation and Pediatric Surgery, University Hospital Schleswig-Holstein, Campus Kiel, Germany. Exclusion criteria were age <18 years, hyperthyroidism and iodine allergy. Patients who were not considered eligible to undergo robot-assisted liver surgery were also excluded.

### 2.3. ICG Application

One vial of ICG (25 mg, Verdye^®^, Diagnostic Green GmbH, Aschheim, Germany) was dissolved in 50 mL sodium hydrochloride or water for injection according to the operator’s manual and applied intravenously immediately after dilution the day before surgery.

### 2.4. Surgical Procedures Using the FireFly^TM^ Camera

The da Vinci Xi^®^ Surgical System (Intuitive, Sunnyvale, CA, USA) was used for all procedures in a standard fashion. All patients were operated on in a supine position. Five trocars were placed according to the tumor location. Usually, two 12 mm trocars and three 8 mm trocars were used. After the first entry, capnoperitoneum was established and maintained throughout the operation. After primary visual inspection of the abdominal cavity, with special attention to the liver, the ultrasound probe was inserted and the liver thoroughly examined by IOUS. Intraoperative findings were correlated with preoperative images. IOUS was repeated on demand during the procedure [5]. 

The Firefly^TM^ camera is integrated into the da Vinci Xi^®^ Surgical System, and is easily switched on and off by pushing a button. It is integrated into the normal camera, so that the NIR image is projected onto the normal camera image. The NIR image appears within seconds and the surgeon at the robotic console does not have to change monitors or vision, but can continue to perform surgery as planned while keeping their eyes on the operation field. ICG accumulation appears green on the screen, while the rest of the operation field appears in different shades of gray. A hybrid of the normal image with the NIR image can also be established. The NIR light can be activated and used on demand during the procedure to observe ICG enhancement. Three different types of ICG accumulation have been described by Ishizawa et al. [19]: Fully fluorescent, partly fluorescent, and the rim type. Different fluorescence patterns were attributed to impaired cellular excretion mechanisms, resulting in intracellular ICG accumulation [19]. Well-differentiated HCC therefore show homogenous fluorescence in the whole tumor, whereas dedifferentiated HCC only show partial accumulation [16]. HM, not consisting of hepatic tissue, do not metabolize ICG but compress cells at their rim, thereby hindering ICG excretion and causing the rim phenomena [7,12,17,22]. Furthermore, increased endothelial peritumoral leakage has been reported to contribute to the rim phenomenon [29]. In cirrhotic liver tissue, ICG accumulation may be less obvious.

Instruments usually used are curved Tip-Ups, monopolar curved scissors, Harmonic Ace^®^ Curves Shears (Intuitive, Sunnyvale, CA, USA), and fenestrated bipolar forceps. Depending on the localization of the lesion, liver mobilization was realized afterwards and the tumor or metastasis, respectively, resected using a crush clamp technique. If larger vessels were near the resection margin or the lesion, Hemolok clips were applied. After resection, the Firefly^TM^ camera was used to verify whether all potentially malignant tissues had been resected. The specimen was usually removed via one of the 12 mm incisions in a recovery bag. If necessary, an enlargement of one incision was performed. If thought to be useful, a drain was placed. Before closure and release of the capnoperitoneum, the resection margin and the whole abdominal cavity were checked for hemostasis using the Valsalva maneuver. Capnoperitoneum was released and standard abdominal closure performed. Patients were extubated immediately after surgery and usually returned to the normal ward after a short period in the recovery room.

### 2.5. Outcome Measures

The primary outcome measure was the feasibility and safety of preoperative intravenous ICG application. The secondary outcome measure was the intraoperative use of preoperatively-applied ICG and its advantages while performing R0 liver-sparing robot-assisted liver surgery.

### 2.6. Statistical Analysis

Qualitative data are presented as means ± standard deviations (SD) and ranges. Quantitative data are presented as percentages. Survival data were analyzed and interpreted using the Kaplan–Meier method [30]. Survival was defined from surgery to last contact or death, whichever occurred first. GraphPad Prism was used to present data. Statistics were performed using GraphPad Prism Version 8 and Microsoft Excel for Mac.

## 3. Results

During the study period (February 2019—November 2020), 147 patients underwent hepatic surgery at the Department of General, Abdominal, Transplant, Thoracic, and Pediatric Surgery, University Hospital Schleswig-Holstein, Campus Kiel. Twenty-seven patients were considered eligible for robot-assisted surgery and received ICG preoperatively, as described. No adverse events or allergic reactions occurred during or after ICG application. Robotic surgery was initiated as planned. Due to anatomical circumstances, the size of the tumor, or proximity to hepatic veins, conversion to open surgery was necessary in six cases (22.2%). One robotic procedure in a highly overweight patient (BMI = 38.1 kg/m^2^) was stopped since the tumor located in segment VIII was deemed too difficult to expose and forced surgical resection would have caused more harm to the patient (who later underwent successful liver transplantation). ICG was not tested in these patients and they were excluded from the final analysis.

In total, 20 patients were included in the final analysis. The mean age was 64.0 ± 12.3 (40–82) years, and more patients were male (55.5%) and suffered from being overweight (median BMI 27.4 ± 6.7 (19.5–40.4) kg/m^2^) (Table 1). HCC was the main diagnosis for surgery (*n* = 5), followed by hepatic metastasis of colorectal cancer (*n* = 5) and neuroendocrine tumor, breast cancer, focal nodular hyperplasia (*n* = 2 each), choroid coat tumor, esophageal and bladder cancer, and suspicion of cancer (*n* = 1 each) (Table 2).

The duration of surgery was 159.8 ± 72.3 min. However, one patient received a complete robot-assisted proctocolectomy with a J-Pouch and protective stoma simultaneously to the atypical liver resection, prolonging the duration of surgery. If docking times were excluded, robotic surgery took 106.4 ± 40.7 min. Most tumors were located in segment VI, VII, and VIII. Three minor and one major hemihepatectomies and 16 atypical liver resections were performed. The average tumor size was 23.8 ± 11.5 (1–34) mm. In most patients, multiple segments were involved in the resection (Figure 1). These included patients suffering from metastasis of colorectal cancer (*n* = 4), hepatocellular carcinoma (*n* = 3), neuroendocrine tumors (*n* = 2), and urothelial carcinoma, choroid coat melanoma, and breast cancer (*n* = 1, each). No intraoperative complications occurred, and intraoperative blood loss was neglectable. Drains were placed in seven cases in patients with larger resection volume, or who were suffering from severe fibrosis or cirrhosis, or were undergoing an additional surgery within the same procedure. The length of stay (LOS) was 6.6 ± 5.4 days (median 5; range: 2–26) (Table 2). One patient suffered from postoperative hepatorenal failure and exacerbation of their chronic obstructive pulmonary disease. This prolonged LOS to 26 days. Another patient simultaneously underwent proctocolectomy and received a transarterial chemoembolization of parts of their liver remnant, which lengthened the LOS to 15 days. Minor complications were noted in two patients. Major complications occurred in one patient suffering from hepatorenal failure and exacerbation of chronic obstructive pulmonary disease after surgery. 

Histopathological analysis confirmed preoperative suspicion in 85.0% of patients. In 15.0%, only necrotic tissue was detectable (Table 2). Two patients, one suffering from breast cancer and one from esophageal cancer, had been treated with chemotherapy preoperatively, explaining the necrotic tissue in two histopathological analyses (Table 2). Histopathologically-proven R0 resections were achieved in 85.0% of patients. All patients with R0 resections showed good intraoperative ICG staining. In one case, an extension was performed after a persistent ICG signal after lesion resection, achieving an R0 situation. ICG, combined with IOUS, was considered most helpful by the surgeon performing the procedure (Table 3). Two patients with an R1 situation suffered from progressive liver cirrhosis and did not show a helpful ICG signal (Table 4). All R1 situations were noted in patients showing unsuccessful ICG staining. Additionally, it is noteworthy that ICG did not help in patients who were mostly older than 70 years, suffering from different types of hepatic metastasis, and mainly suffered from liver cirrhosis and fibrosis, respectively (Table 4).

The mean follow-up period was 9.4 ± 6.7 months. Cancer-specific survival was 100%, while the overall survival was 94.7%. The patient with the prolonged postoperative course died three months after surgery after a fall in a rehabilitation clinic. Recurrence-free survival was 8.7 ± 0.5 months. Recurrence occurred after a mean of 10.4 ± 2.6 months in seven patients (36.8%). Recurrences were newly-diagnosed metastasis of neuroendocrine tumor (lymph node), breast (hepatic), esophageal (brain), and colorectal cancer (hepatic), and choroid coat melanoma (ubiquitously). Patients with hepatic recurrence underwent ICG-based surgery again. Three patients without systemic metastasis successfully underwent liver transplants during their postoperative courses. However, due to the heterogeneity of patients included, survival analyses have limited meaning.

The ICG was administered on average 21 h 24 min ± 4 h 52 min (range: 7 h 39–47 h 05) before surgery. The amount of ICG used was on average 0.32 ± 0.08 (0.22–0.50) mg/kg (Table 2). Intraoperatively, ICG accumulation was obvious in 12 cases and correlated with the preoperative imaging and/or the IOUS (Table 3). As previously described by Ishizawa et al., we observed full fluorescence in HCCs, independent of grading, and rim phenomena in the other metastases [16] (Figure 2 and Figure 3). In one case, ICG revealed one additional lesion that had been overlooked by preoperative imaging. In another case, two additional lesions (1 and 2 mm) were resected because of ICG staining. Histopathological analysis also revealed metastatic tissue. In seven cases, however, ICG application did not help as it either did not accumulate, stained the whole liver, or accumulated ubiquitously in the cirrhotic liver (Table 4). One patient received ICG preoperatively, but as the tumor was clearly detectable macroscopically, the FireFly^TM^ camera was not used. Interestingly, ICG showed accumulation in a metastasis of esophageal cancer. However, the postoperative histopathological analysis could only provide proof of necrotic tissue.

## 4. Discussion

This prospective, single-center study reports on our experience regarding the feasibility and intraoperative use of intravenous ICG application in patients undergoing robot-assisted hepatic resections for HCC and HM of different tumors. In our experience, preoperative intravenous ICG application is easy to establish, simple and feasible, without complications, and serves as an additional supportive, real-life intraoperative aid. In most cases, it may help the operating surgeon to differentiate cancerous tissue from normal hepatic tissue by means other than conventional preoperative imaging and IOUS, and thereby help with intraoperative strategic planning.

Due to in-house logistics, all included patients received 25 mg ICG dissolved in 50 mL NaCl or water for solution the day before surgery. Adapted to body weight, the patients received 0.22–0.50 mg/kg between 7:39–47:05 h before surgery (Table 3). The average time difference between application and surgery was 17:26 h. We noticed that for the patient who only received ICG 7:39 h before surgery, the whole liver showed green staining and ICG application did not help at all to perform surgery. The contrary was noted for the patient who had to be rescheduled to the next day due to in-house emergency surgery and therefore received ICG 38:05 h before surgery. Another patient received ICG 47:05 h prior to surgery as he had to undergo an additional preoperative CT scan two days before surgery. In both cases, the ICG signal was unequivocal and helped to perform liver-sparing, straightforward surgery.

The dose and time of ICG application is much discussed in the literature. Alfano et al. considered 0.5 mg/kg applied 24–48 h before surgery ideal to achieve reliable intraoperative staining [22]. Yet other reports regarding time of application ranged from 12 h to 10 days before surgery [15,18,22,23,24,25,26]. If applied too close to surgery, the false-positive rate of ICG accumulation can be quite high, leading to false, unnecessary resections [7,14,16]. Therefore, most authors recommend application intervals of 24–48 h [22,25]. Peyrat et al., even warranted 48–72 h between application and surgery, whereas van der Vorst et al., preferred 72 h if possible [25,26]. The advantage in this setting, unlike intraoperative identification of bile leakage or blood supply of a liver segment, is that patient-specific metabolic and physical properties such as heart rate and blood pressure, do not need to be considered, and a longer interval between application and surgery seems to be favorable [31].

Nonetheless, in fibrotic or cirrhotic liver tissue, hepatic metabolism is impaired. Therefore, impaired and slower hepatic elimination of ICG may lead to false positive or no results as we observed in our study [22,31]. In two patients with liver cirrhosis, the suspicious area or potential HCC, as well as the whole cirrhotic liver tissue, showed staining and small accumulations, respectively, which did not help to differentiate cirrhotic and tumorigenic liver tissue. Additionally, IOUS was difficult to perform in these patients. Perhaps one way to overcome this false positive staining may be a longer interval between ICG application and surgery [7,19]. Initially, Ishizawa et al. advocated at least a 48-h interval, especially in patients suffering from liver cirrhosis, before performing surgery [16]. Kawaguchi et al. even suggest an interval of seven days between ICG application and surgery in patients suffering from liver cirrhosis [27]. Nevertheless, no clear recommendation regarding perfect time point for ICG application for patients with normal or with cirrhotic liver tissue exists.

ICG has few reported side effects, and the lethal dose is estimated between 50–80 mg/kg [23,32]. The standard clinical dose used is between 0.1–0.5 mg/kg [7]. However, doses vary from indication to indication. In the setting of staining hepatic tumors and metastases, dose and timing are key to the avoidance of background fluorescence [25]. Kobayashi et al., like Alfano et al., performed titration experiments with different doses and concluded that either 3.75 mg ICG or 0.2 mg/kg, respectively, were favorable [15,22]. Moreover, Sucher et al. recommended a dose of 2.5–5 mg if only hepatic lesions needed to be visualized [7]. Van der Vorst et al. applied either 10 mg 24 h before surgery or 20 mg 48 h prior to surgery to obtain a clear contrast between tumor and normal liver tissue [25]. They thereby achieved concentrations between 0.13–0.26 mg/kg prior to surgery [25]. We applied an average dose of 0.33 mg/kg, which is a little above most recommendations. This, in combination with the relatively short time of application, may explain the ubiquitous staining in some of our patients. We also noticed that in general, ICG did not provide additional information in elderly patients with slower metabolisms. Some also suffered from liver cirrhosis. If the duration between application and surgery is short, smaller doses of ICG injection seem to be favorable, leading to fewer false positive results [7,25]. We would therefore rather stick to reduced doses and try to apply ICG a longer time (i.e., 48 h rather than 24 h) before surgery.

Parenchymal-sparing R0 resection is key to prolonged survival in oncological surgery [24,25]. Preoperative work-up—consisting of ultrasound, computed tomography (CT), and magnetic resonance imaging (MRI) scans—therefore aims to identify as many lesions as possible to plan surgery accordingly and achieve the oncological best situation [12,23]. Nonetheless, some lesions, especially small, superficial ones, may be overlooked in pre- or intraoperative conventional imaging, thereby highly influencing the oncological prognosis [6,12,16,24,25,27,33,34]. After ICG application, substantially higher detection rates of additional lesions and even primary detection due to intraoperative ICG staining have been reported [14,17,18,23,25,26]. Kudo et al., identified another 17 lesions in 17 patients, whereas van der Vorst reported detection of additional metastases in five patients [12,25]. We can report ICG-based identification of additional nodules, i.e., in the patient suffering from choroid coat melanoma metastasis and one patient suffering from metastasis of a neuroendocrine tumor. In both cases, preoperative imaging and IOUS had not shown the additional foci, but intraoperative ICG did. Boogerd et al., even reported clear superiority of ICG in detecting nodules compared to preoperative CT, MRI, and IOUS [28]. Besides being highly user-dependent, IOUS has the problem of not detecting lesions that are just below the surface within the first cm of the liver [16,24]. In contrast, NIR light can only penetrate up to 1 cm into liver parenchyma and thereby fails to detect deeper tumors [18,33]. The combination of IOUS and ICG therefore seems to increase the detection rate of hepatic metastasis [31].

Along with a higher detection rate, Handgraaf et al., reported better survival after ICG-orientated liver resections due to the resection of additional nodules, which had been missed by pre- and intraoperative imaging [34]. Accordingly, Marino et al., compared robot-assisted liver resections with and without additional ICG application and reported significantly higher R0 resection rates after ICG application [17]. In our cohort, ICG staining helped to perform a R0 resection in one patient. After resection of the ICG-stained, ultrasonographically-verified lesion, the resection margin still showed ICG staining and an extended resection was immediately performed, finally achieving an R0 situation. Due to the visualization of ICG accumulation in around 200 cells, small lesions—which otherwise would have been missed—can be detected [16,31]. However, oncological long-term results comparing ICG-based surgery with conventional surgery have not yet been published [7].

Nevertheless, some authors have already called for mandatory preoperative ICG application in addition to IOUS, as each method seems to complement the other; in contrast, other experts suggest that it should only be used as an additional aid [16,18,24,25,26,27,33,34,35]. In our experience, ICG application can be easily implemented into daily practice at low cost, providing additional real time information during robotic surgery when haptic feedback is missing [23]. Furthermore, visualization of ICG accumulation can be shown continuously without having to change instruments or monitors during surgery—a great advantage over IOUS, for example, and helping to improve intraoperative navigation [7,23]. Therefore, ICG presently seems to be a helpful additional aid in robot-assisted liver surgery, making tailored liver parenchyma-sparing liver resections possible [18]. However, to our knowledge only two reports on the use of ICG in robotic surgery have been published, even though they considered preoperative ICG application to be useful [17,18]. 

The main limitation of our study is its design as a single-center, single-arm feasibility study that only included a small number of unselected patients suffering from different types of cancer, who received ICG at different doses and different points of time. Unfortunately, we cannot provide data for a comparative cohort. Additionally, ICG specific limitations must be considered, including low penetration depth (up to 13 mm), its non-quantifiable nature, and lack of reliability in patients suffering from liver cirrhosis [14,16,33]. Until now, no standardization of the intensity of fluorescence has been established [16,19,32]. Furthermore, the location of tumors and metastases, respectively, appear to be problematic due to the short penetration depths of NIR light.

## 5. Conclusions

Intraoperative hyperspectral imaging, artificial intelligence, deep machine learning, augmented reality, fusion of preoperative CT- or MRI-scans to IOUS and the operation field, and other technical developments will be possible in the near future using the robotic system, enabling new perspectives in hepato-biliary surgery and hopefully achieving better oncological results and patient survival [1,4,8,22,32,36,37,38,39,40]. Until then, ICG staining may serve as an additional helpful intraoperative aid in patients undergoing robot-assisted, atypical liver resection, providing real-time images and helping to plan intraoperatively and perform surgery in an individual, tissue-sparing way. In our experience, ICG is not only applicable to HCC and CRC metastasis, but also works for HM of other tumors. However, the operating surgeon should not only rely on ICG staining but integrate it into the repertoire of intraoperative planning tools. Further studies are needed to determine the exact dose and time of application and standardize fluorescence intensity. It would also be helpful to have a dye that specifically targets or accumulates in cancer cells.

## Figures and Tables

**Figure 1 jcm-10-00456-f001:**
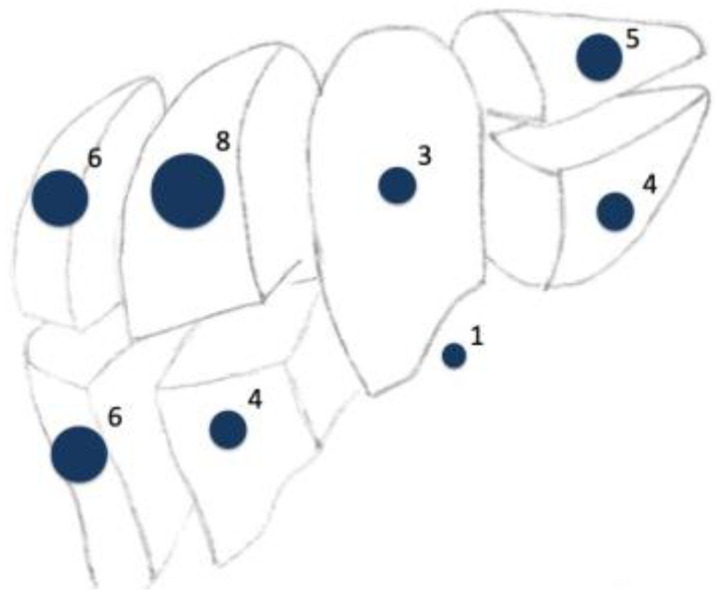
Scheme of location of tumors and metastasis resected.

**Figure 2 jcm-10-00456-f002:**
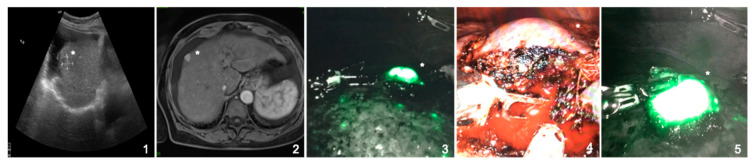
Pre- and intraoperative imaging of a 64-year-old female patient suffering from a hepatocellular carcinoma in liver cirrhosis (CHILD A). Indocyanine green accumulation in the hepatocellular carcinoma is clearly visible and shows full fluorescence (*). (**1**,**2**) Preoperative ultrasound and MRI-scan showing a hepatocellular carcinoma in segment VIII; (**3**) intraoperative, near infrared Firefly imaging showing indocyanine green accumulation in the tumor and the cirrhotic liver; (**4**) naive intraoperative image of the tumor; (**5**) intraoperative, near infrared Firefly imaging of the resected tumor.

**Figure 3 jcm-10-00456-f003:**
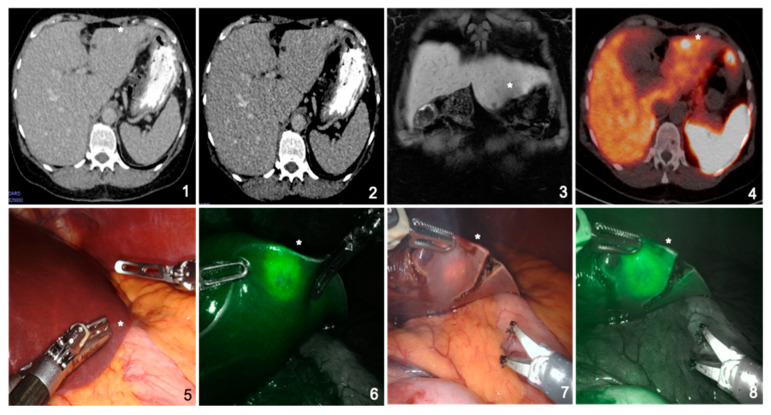
Pre- and intraoperative imaging of a 58-year-old female patient suffering from hepatic metastasis of a neuroendocrine tumor in a noncirrhotic liver. Indocyanine green accumulation at the rim of the metastasis is clearly visible and shows rim fluorescence, even after partial resection (*). (**1**–**3**) Preoperative computed tomography and magnetic resonance imaging scans do not show obvious metastasis; (**4**) preoperative DOTATATE-PET revealing subsuperficial metastasis in segment III (*); (**5**) naive intraoperative image not giving a macroscopic hint of the metastasis (*); (**6**) intraoperative, near infrared Firefly imaging showing indocyanine green accumulation at the rim of the metastasis; and (**7**,**8**) intraoperative, near infrared Firefly imaging showing indocyanine green accumulation at the rim of the partly resected metastasis (*).

**Table 1 jcm-10-00456-t001:** Demographic data of patients.

	(*n* = 20)
Age (years), mean ± SD (range)	64.0 ± 12.3 (40–82)
Gender, % male	55.5
BMI (kg/m^2^), median ± SD (range)	27.4 ± 6.7 (19.5–40.4)

BMI, body mass index; SD, standard deviation.

**Table 2 jcm-10-00456-t002:** Surgery- and tumor-specific data.

	(*n* = 20)
Time of surgery (min), mean ± SD (range)	159.8 ± 72.3 (75–363)
Time of console (min), mean ± SD (range)	106.4 ± 40.7 (34–315)
Histopathological result (preoperative diagnosis/final diagnosis), *n*	
Hepatocellular carcinoma	7/5
Colorectal cancer	5/5
Neuroendocrine tumor	2/2
Breast cancer	2/1
Follicular nodular hyperplasia	1/2
Choroid coat melanoma	1/1
Urothelial carcinoma	1/1
Esophageal cancer	1/0
Unspecific	0/3
Size of tumor (mm), mean ± SD (range)	23.8 ± 11.5 (1–34)
Postoperative complications (Clavien-Dindo I/II/III/IV), *n*	2/0/0/1
Length of hospital stay (days), mean ± SD (range)	6.6 ± 5.4 (2–26)

*n*, number of patients; SD, standard deviation.

**Table 3 jcm-10-00456-t003:** Intraoperative indocyanine green (ICG) use (according to surgeon’s perception).

	(*n* = 20)
Dose of ICG applied (mg/kg) mean ± SD (range)	0.32 ± 0.08 (0.22–0.50)
Duration between ICG application and surgery (h:min) mean ± SD (range)	21:24 ± 4:52 (7:39–47:05)
Intraoperative ICG signal *^a^*	2.4 ± 1.4 (1–6)
The intraoperative ICG signal helped during surgery (yes), % (*n*)	60.0 (12)
ICGA was clear and unequivocal (yes), % (*n*)	60.0 (12)
IOUS was used (yes), % (*n*)	100 (20)
Did IOUS and ICGA correlate? (yes), % (*n*)	75.0 (15)
Which intraoperative support helped the most? *n*	
ICGA	3
IOUS	4
Combination	8
None	4
None necessary because of macroscopic detection	1

ICG, indocyanine green; ICGA, ICG accumulation; IOUS, intraoperative ultrasound; n, number of patients. *^a^* 1 = excellent signal, 2 = good signal, 3 = moderate signal, 4 = sufficient, 5 = insufficient signal, 6 = no signal at all.

**Table 4 jcm-10-00456-t004:** Patients showing unsuccessful ICG staining.

	1	2	3	4	5	6	7
Sex	m	f	f	m	f	m	f
Age (years)	61	70	40	76	74	79	82
Comorb. (liver)	Cirrhosis (grade IV)	cirrhosis	—	fibrosis	—	—	cirrhosis (grade IV)
Dose (mg/kg)	0.23	0.50	0.28	—	0.30	—	0.32
Timediff (h:min)	—	18:25	—	15:32	7:39	12:41	17:31
IOS	df	us	ns	ns	ns	us	ns
Histopath (Resection margin)	HCC (R1)	HCC (R1)	BC	HCC	NM, initally BC	UC (R1)	NM

BC, breast cancer; Comorb., liver-associated liver comorbidities; df, diffuse staining; HCC, hepatocellular carcinoma; Histopath., final histopathological results; IOS, intraoperative staining; NM, no malignancy; ns, no staining; Timediff, time difference between application and surgery; UC, urothelial carcinoma; us, ubiquitous staining.

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
