# Peer review of "Usability of Indocyanine Green in Robot-Assisted Hepatic Surgery"

_jcm, 2021, doi:10.3390/jcm10030456_

Round 1

Reviewer 1 Report

Dear Authors, 

I carefully the paper that sounds very interesting. I have some minors recommendations/point I would like you address: 

  • The introduction seems to long, I will skip the part related to the metabolism/application of ICG 
  • Line 82 the sentence "the ICG is mainly used to test liver function especially in Asian countries" requires a citation or maybe it is better to erase it
  • Inclusion criteria: did you include only liver mets or HCC, so do you routinely perform a liver biopsy during the work-up?
  • Line 119 "ultrasound electrode" stands for "ultrasound probe"?
  • Line 121 correct as "Tip-Up"
  • Material and Methods can you please mention the exclusion criteria from the study?
  • Line 150-153, not clear, I suggest to erase this sentence
  •  Can you please comment some data of the table 3, ICGA was clear and unequivocal or the ICG signal helped during surgery; did the surgeon evaluate or did he complete a questionaire?
  • Can you also explain why the timing of ICG injection varies during the study, it is repaired as 17 ± 8 hours

Author Response

Dear Reviewer 1, 

Thank you for considering our revised manuscript entitled "Usability of Indocyanine Green in Robot-Assisted Hepatic Surgery" for re-submission at the Special Issue on "Recent Advances in Minimally Invasive Surgery".

Firstly, we would like to thank you for reading through our manuscript and for the excellent and helpful remarks. We appreciate your comments and welcome the opportunity to address these queries and concerns in our revised manuscript. We have addressed all your comments in our responses. 

We hope the revised manuscript, together with our point-by-point discussion, answers all questions and makes in suitable for publication in this special Issue on "Recent Advances in Minimally Invasive Surgery".

Yours Sincerely, 

Jan-Hendrik Egberts

Reviewer 2 Report

Review comments:

Language:

The article contains numerous errors in spelling and in the use of the English language. It would help to have it spell checked by a native speaker.

The general opinion is that the article suffers from a lack of clarity.

Although it is a very descriptive article, there has to be a clearer description of the objective of this manuscript.

Introduction

General comments:

- Please clearly state the study goal, this remains unclear after reading the introduction (also after reading the abstract).

Lines 33-35: Advantages of the robotic platform include smaller incisions, clearer visualisation 34 of structures, higher degrees of freedom, avoidance of the fulcrum effect and better access to 35 segments IVa, VII and VIII

Compared to laparoscopic surgery?

Line 44-45: The surgeon has to rely on his or her visual impressions

Line 46-47: ICG is not a tool itself

Line 50: year of FDA approval should be 1957

Lines 72-74: this is true for open procedures but untrue for laparoscopic surgery.

Methods

General comments:

- The surgical procedure is extensively described, but does not relate to the purpose of the study, herein: ICG imaging for hepatic malignancies

- Outcome measures: please further specify with scientific terms. 'ICG-based robot-assisted hepatic resection' cannot serve as a study endpoint. Neither can 'use of ICG' serve as a secondary endpoint. The authors might refer to study the safety and feasibility of ICG.

- I am not certain if a survival curve is suitable for a 19 patient study with a very heterogenous patient group, with different disease characteristics.

Line 111: acqua? Water for injection

Results

General comments:

Length of hospital stay varies widely in the results. How did the LOS correlate to type of surgery (i.e. minor hepatectomy/major hepatectomy)? Why were drains placed if there were no intraoperative complications and blood loss was negligible in all cases?

Line 155-156: bring sub sentence “overweight (BMI…)” forward.

Lines 157-160: number of patients per preoperative diagnosis do not correspond to the table

Lines 180-181: unclear results, where is the last 10.5%? Were the patients with only necrotic tissue pretreated with chemotherapy?

Lines 208-210: Interesting; images available?

Lines 213-214: no rim-shaped enhancement? Image available?

Discussion

General comments:

First paragraph is not in line with the results. In 7/19 patients ICG staining was unsuccessful.

Line 245: 'aqua' is not a scientific term. 'Water for solution' or 'demineralized water' are more appropriate

Lines 250, 292, 348: spearing = sparing

Lines 304-309 … foci, but intraoperative ICG did. Boogerd et al. even reported on a clear superiority of ICG in detecting nodules compared to preoperative CT, MRI and IOUS [31]. Besides being highly user dependent, pre- but also intraoperative IOUS has to face the problem of missing out on lesions, which are just underneath the surface within the first cm of the liver [15,27]. On the contrary, NIR-light can only penetrate up one cm (I think it is less) into liver parenchyma and thereby fails to detect deeper located tumours [17,34]. The combination of IOUS and ICG seems to increase the detection rate of hepatic metastasis [24].

Conclusion

Lines 349-350: reports exist describing the use of ICG in liver metastases of other origin than CRC.

Tables: Table 1-3 do not provide the overview a table should provide, this might be an easy fix by alignment of the text to the left.

Table 1.

Please use commas to divide the descriptions of abbreviations

Table 2.

Describe difference in preoperative diagnosis and histology

Table 3.

What does the asterisk refer to?

non – change in None

Please use commas to divide the descriptions of abbreviations

Table 4.

What does 'W' stand for? An 'F' for female might be more appropriate

What was the initial diagnosis of patient 7? And what did histology show?

Figures.

Figure 1

Figure suggests 35 lesions were found. Specify which patients had multiple lesions.

Figures 2 and 3

- Image 3,5 (fig 2) 10 and 12 (fig 3) are not suitable for publication: image 3 and 5 suffer from major over-exposition of NIR light, and image 10 and 12 suffer from autofluorescence from (presumably) surrounding lights from the operation theatre and are therefore not representative. Image 6 and 8 (figure 3) are very clear examples of in vivo NIRF imaging with ICG and should be the standard.

Author Response

(The authors gave the same response as above.)

Reviewer 3 Report

Dear Editor and authors,

I would like to emphasize the promising nature of the study proposed by Medhorn AS et al.

Medhorn SA et al. present an observational study of 19 cases showing the usefulness of ICG fluorescence during robot-assisted liver resection.

The use of ICG fluorescence during laparoscopic and open hepatectomy has already been widely published. Recently, the use of the robot-assisted approach for liver resection has been described. Thanks to The Firefly TM camera integrated in the DaVinci Robotic Systems (Intuitive, California, USA), this camera can easily be used intraoperatively visualizing ICG accumulation.

The small size of the study; the absence of comparative analysis of ICG versus no ICG for robot-assisted liver resection; the absence of comparative analysis of laparoscopic versus robot-assisted liver resection using intraoperative ICG fluorescence, confer to the present study a low level of scientific evidence.

In the present study, Medhorn AS et al. reported the efficacy of ICG fluorescence in 11 patients (57.8%). In contrast, Table 4 reports only 7 patients in whom the use of ICG fluorescence was not convincing. One case is therefore not explained. Furthermore, the results show that ICG fluorescence seems less helpful than IOUS. One of the hypotheses may be the histological diversity of the lesions operated on, which may have a different behavior on ICG accumulation after injection, for the same preoperative injection protocol. Indeed among the 7 ICG failures, 4 were neither HCC nor colorectal cancer liver metastases.

Recently, Marino MV et al. have already published two studies in 2019 and 2020 on the usefulness of ICG during robot-assisted liver resections, with a clear message. In 2019, Twenty-five patients who underwent ICG fluorescence-guided robotic liver resection (HCC and colorectal cancer metastases) were case-matched in a 1:1 ratio to a cohort who underwent standard robotic liver resection. Despite the similar operative time (288 vs. 272 min, p = 0.778), the risk of postoperative bile leakage (0% vs. 12%, p = 0.023), R1 resection (0% vs. 16%, p = 0.019) and readmission (p = 0.023) was reduced in the ICG group compared with the no-ICG group. In 2020, the authors have analyzed the impact of ICG fluorescence staining technique in 40 consecutive patients who underwent robotic-assisted liver resection for malignancies.

In these two studies, the tumor detection rate of intraoperative ICG was more than 85%. They additionally performed ICG fluorescence detection of the resected specimens after removal and found a detection rate of 100%. While the ICG fluorescence of liver surface detected additional tumors than intraoperative ultrasonography, it failed to detect other tumors confirmed by the fluorescence imaging of the resected specimens. All missed tumors had similar mean diameter compared to the identifiable tumors but were located deeper) into the liver parenchyma. In the study of Medhorn SA et al. another reason that may explain the ineffectiveness of ICG, may be the deep localization of lesions.

Author Response

(The authors gave the same response as above.)

Round 2

Reviewer 2 Report

Although a lot of revisions have been made by the authors, the main concern is the lack of originality (for example: Marino MV et al. World J Surg. DOI: 10.1007/s00268-019-05055-2). The authors could rewrite the manuscript to a center experience article with or without a literature review.

Reviewer 3 Report

Dear,

This is a promizing study.

However, assessing the usability of indocyanine green in Robot-Assisted hepatic surgery his not new. Futhermore the small size (n=20) of the present study with only 58% of ICG efficacy adress to the present work a low degree of scientific evidence.

Additionnal experiment are needed.